# The attitude of kidney transplant recipients towards elective arteriovenous fistula ligation

**Klaudia Bardowska**[1], **Krzysztof Letachowicz**[2]*, **Dorota Kamińska**[2], **Mariusz Kusztal**[2], **Tomasz Gołębiowski**[2], **Tomasz Królicki**[1], **Karolina Zajdel**[1], **Oktawia Mazanowska**[2], **Dariusz Janczak**[3], **Magdalena Krajewska**[2]

**1** Faculty of Medicine, Wroclaw Medical University, Wrocław, Poland, **2** Department of Nephrology and Transplantation Medicine, Wroclaw Medical University, Wrocław, Poland, **3** Department of Vascular, General and Transplantation Surgery, Wroclaw Medical University, Wrocław, Poland

* krzysztof.letachowicz@umed.wroc.pl

## Abstract

### Background

Arteriovenous fistulas (AVF) are a source of various complications. Among previously hemodialyzed kidney transplant recipients (KTxR), the AVF may persist over time. The patients' decisions whether to ligate the functioning AVF may be prompted by many factors. Our knowledge of benefits concerning the procedure as well as patients' attitude towards it is scarce.

### Aim

Evaluation of the patients' opinion on the persistent AVF ligation after a successful kidney transplantation.

### Materials and methods

An anonymous survey was carried out among 301 previously hemodialyzed KTxR. The patients were recruited during scheduled visits in the Transplantation Outpatient Unit. All subjects completed an anonymous questionnaire including questions about their attitude towards the matter in question.

### Results

69 patients (22.9%) have considered AVF closure. The most common causes for such attitude were esthetic reasons (n = 29) and concerns about heart health (n = 13). Among those 69 subjects, 18 have presented with symptomatic AVF due to multiple symptoms. Symptomatic AVFs were localized on the forearm in 14 out of 18 cases. As many as 116 (38.5%) cases have never wanted to ligate the AVF and 116 (38.5%) subjects did not have a clear opinion. In our study we report 158 (52.5%) cases of non-functioning AVFs. The main reason for the above was spontaneous AVF thrombosis (121 cases). Only 24 subjects reported to rely on the physician-provided information about the AVF management.

**Data Availability Statement:** The data are available at https://doi.org/10.17632/rppgbwzzgs.2

**Funding:** The study is supported by the Wroclaw Medical University statutory funds (SUB.

C160.19.055). It was an investigator initiated research. The funding body had no role in study design, data collection, analyses, and interpretation, or in writing the manu-script.

**Competing interests:** The authors have declared that no competing interests exist.

## Conclusions

One fourth of KTRs have ever considered AVF ligation. There is a distinct need for educating patients on the possibilities of post-transplantation AVF management.

## Introduction

Nowadays, kidney transplantation is the first line treatment in the end stage renal disease (ESRD) due to better clinical outcomes, enhanced quality of life (QOL) and its higher cost-effectiveness compared to other renal replacement therapies [1–4]. The intensive growth of multiple national as well as international transplantation programs has led to the point where both hemodialysis and peritoneal dialysis have become bridging-therapies during which the patients are waiting for a suitable kidney donor. Appropriate medical care, including regular follow-ups, an immunosuppressive regimen and early acute rejection treatment have also made it possible to obtain low graft loss rates of around 3% per year [5]. As hemodialysis remains a predominantly applied dialysis method (86.9% of newly initiated dialysis) in USA in 2017 [6], the number of previously hemodialyzed kidney transplant recipients with persistent arteriovenous fistula (AVF) systematically rises in the transplant recipient community. As a result, concerns have been raised about proper management of these patients, as in most cases the AVF remains redundant.

Education of hemodialyzed patients about proper AVF maintenance is a cornerstone to ensure its patency and thereby hemodialysis feasibility [7,8]. Several studies have already been focused on the patients' attitude towards an already created AVF as well as their reluctance towards vascular access creation [9,10]. However, little is known about the attitude of kidney transplant recipients (KTRs) towards persistent AVF. The aim of the study was to evaluate patients' opinion on persistent AVF ligation after the successful kidney transplantation.

## Materials and methods

### Study design

The participants were recruited from among 400 consequent kidney transplant recipients who had a scheduled follow up appointment in the Transplantation Outpatient Unit of the university center in Poland. All subjects completed an anonymous survey (S1 File), with the response rate of 88.25% (n = 353). In this group, 301 patients were found to be eligible for this study. The study was being carried out from March 1, 2019 to April 30, 2019. Raw data set was deposited in a repository and is available at https://doi.org/10.17632/rppgbwzzgs.1.

### Inclusion criteria

- Adult outpatient kidney transplant recipient (aged 18 or more)

- Hemodialysis preceding transplantation with the use of AVF

### Questionnaire

The questionnaire included questions regarding patients' basal characteristics: gender, age, anthropometric data, dialysis type and duration time, current serum creatinine concentration,

comorbidities, information about presence, localization and patency of AVF. All patients answered a series of 5 questions:

1) Have you ever considered AVF ligation and why? (YES / NO / I DO NOT KNOW)

2) Has the following influenced your attitude? (physician suggestion/ family suggestion/ esthetic reasons)

3) If the AVF is not active, what is the cause? (ligation by physician/ thrombosis)

4) When did the ligation / thrombosis occur?

5) Has your condition changed after cessation of AVF function? (Yes—I felt better / Yes—I felt worse / No)

## Statement of Ethics

The research was conducted ethically and in accordance with the World Medical Association Declaration of Helsinki. The study was approved by the Local Ethics Committee of Wroclaw Medical University (approval number KB-775/2018). All patients signed the informed consent form.

## Statistical analysis

Descriptive data are presented as frequencies and percentages. To compare them, chi-squared tests or Fischer's exact test were performed as appropriate. The distribution of continuous variables was assessed with the use of Saphiro-Wilk test. Continuous data are presented as median and interquartile ranges (IQR) due to skewed distribution. The significance of differences between these data was tested using the independent Mann-Whitney U test. A two-tailed P-value of $< 0.05$ was statistically significant. All analyses were performed using Statistica 13.2 (StatSoft, Tulsa, OK, USA).

## Results

In the whole study group (n = 301), 69 patients considered AVF closure, 116 patients negated such considerations and 116 patients did not have a clarified opinion on the given matter. The patients were classified into 2 groups, according to AVF patency: AVF(+) (n = 143) and AVF (-) (n = 158). Table 1 presents the comparison of these groups including: demographic and anthropometric data, characteristics of AVF (localization, cause of malfunction), patients' comorbidities and ESRD cause. The leading cause of non-functioning AVF in the AVF(-) group was a spontaneous thrombosis, which occurred in 76.6% of the analyzed cases, while AVF ligation was performed in 12.7% of the patients. Only 53% (n = 84) of AVF patients could give the precise time of AVF thrombosis or ligation. It should be emphasized that in the AVF(+) group, the vascular access was recreated after thrombosis or ligation of previous AVF in 18 patients. The AVF(+) group showed also a significantly higher serum creatinine concentration (p = 0.0250), however, after post-transplant adjustment, this difference was no longer significant (p = 0.0513). The median time from transplantation was significantly higher in the AVF(-) group. The most common cause of ESRD in both groups was glomerulonephritis, followed by polycystic kidney disease and hypertensive nephropathy.

In the lower section of Table 1, the patients' attitude towards AVF ligation was presented. In the AVF(-) group only 24 subjects considered the ligation of vascular access, which clearly corresponds to the 20 ligation procedures performed in this group. 66.5% of this patient group did not have a clarified opinion on this matter. On the contrary, 31.5% of respondents in the

**Table 1. Characteristics of the study group, according to AVF patency.**

| | AVF+ (n = 143) | AVF- (n = 158) | p-value |
|---|---|---|---|
| Baseline characteristics: | | | |
| Males/Females | 97/46 | 80/78 | 0.0025 |
| Age [years] | 58 (44–64) | 57 (45–63) | 0.5484 |
| BMI [kg/m$^2$] | 25.7 (23.7–28.5) | 25.9 (23.3–29.4) | 0.7941 |
| Serum creatinine [mg/dL] | 1.45 (1.2–1.66) | 1.32 (1.1–1.6) | 0.0292 |
| Dialysis time [months] | 24 (15–41) | 23 (12–36) | 0.0607 |
| Time from transplantation [months] | 84 (42–165) | 162 (79–185) | <0.0001 |
| Time from transplantation to AVF-ligation/-thrombosis*[months] | - | 2.5 (1–31) | |
| Primary vascular access (applies to AVF+) [n,%] | 125 (87.4%) | - | |
| Secondary vascular access (applies to AVF+) [n,%] | 18 (12.6%) | - | |
| Reasons for cessation of AVF function (in AVF+ group applies to a previous vascular access) [n,%]: | | | |
| AVF- ligation | 4 (2.8%) | 20 (12.7%) | |
| AVF- thrombosis | 14 (9.8%) | 121 (76.6%) | |
| Unknown | - | 17 (10.7%) | |
| Leading etiology of CKD [n, %]: | | | |
| Glomerulonephritis | 68 (47.6%) | 80 (50.6%) | 0.5934 |
| Polycystic kidney disease | 24 (16.8%) | 19 (12.1%) | 0.2388 |
| Hypertensive nephropathy | 16 (11.2%) | 13 (8.2%) | 0.3846 |
| Diabetic nephropathy | 7 (4.9%) | 9 (5.7%) | 0.7571 |
| Other | 28 (19.5%) | 37 (23.4%) | 0.4191 |
| Comorbidities [n,%]: | | | |
| Coronary artery disease | 25 (17.5%) | 20 (12.7%) | 0.2411 |
| Heart failure | 26 (18.2%) | 21 (13.3%) | 0.2431 |
| Diabetes mellitus | 26 (18.2%) | 38 (24.1%) | 0.2140 |
| Active smoker | 13 (9.1%) | 7 (4.4%) | 0.1050 |
| History of smoking | 52 (36.4%) | 51 (32.3%) | 0.4557 |
| Localization of AVF [n,%] | | | |
| Distal extremity | 90 (62.9%) | 83 (52.5%) | 0.0505** |
| Elbow area | 29 (20.3%) | 17 (10.8%) | 0.0517** |
| Proximal part of extremity | 14 (9.8%) | 5 (3.2%) | 0.1033** |
| Unknown | 10 (7%) | 53 (33.5%) | |
| Have you ever considered AVF ligation and why? | | | |
| YES: | 45 (31.5%) | 24 (15.2%) | |
| Esthetic reasons | 21 | 8 | |
| I have concerns about heart health | 10 | 3 | |
| Discomfort or pain caused by the AVF | 11 | 3 | |
| Ischemic symptoms of the extremity | 3 | 0 | |
| Inflammation of the AVF | 1 | 0 | |
| The AVF-flow disturbs me in my sleep. | 1 | 0 | |
| The AVF-flow disturbs my wife during sleep. | 1 | 0 | |
| Unknown | 0 | 10 | |
| NO: | 87 (60.8%) | 29 (18.3%) | |
| I would like to preserve my AVF for the future. | 4 | 0 | |
| I do not have a clarified opinion. | 11 (7.7%) | 105 (66.5%) | |
| The AVF feels neutral to me. | 11 | 0 | |
| The influence of third parties on the patients' decisions: | | | |
| Suggestion made by the physician | 15 | 9 | |

(*Continued*)

**Table 1.** (Continued)

| | AVF+ (n = 143) | AVF- (n = 158) | p-value |
|---|---|---|---|
| Suggestions made by the family | 3 | 0 | |

*only 53% of patients from the AVF- group could give the precise date of AVF function cessation.

**p-values calculated after excluding missing data regarding AVF-localization.

AVF(+) group wanted to ligate AVF, 60.8% neglected such considerations and only 7.7% did not have a clarified opinion. The most common reasons given for considering AVF closure were esthetic reasons (n = 29), followed by concerns related to heart health (n = 13). Among patients who expressed their willingness for AVF closure, 24 subjects reported that AVF ligation had been suggested to them by a physician, whereas 3 subjects were advised in this matter by a family member. In the whole study group, 18 cases of symptomatic AVFs were identified and they were located predominantly on the forearm, n = 14 (77.8%).

The attitude of patients with persistent AVF towards its closure was also investigated according to kidney graft function (expressed by serum creatinine levels), time from transplantation and the localization of AVF. The results were presented in Figs 1–3, as appropriate. In Fig 1, patients were divided into 3 groups, according to creatinine concentration: Serum creatinine <1.5mg/dL, 1.5–2.0 mg/dL and >2mg/dL. The highest proportion of patients willing to ligate their AVF was present in the group with the highest creatinine concentration (Fig 1). Additionally, the proportion of these patients rose systematically in time after the KTx, whereas the proportions of patient who did not wish to ligate their AVF sank parallelly (Fig 2).

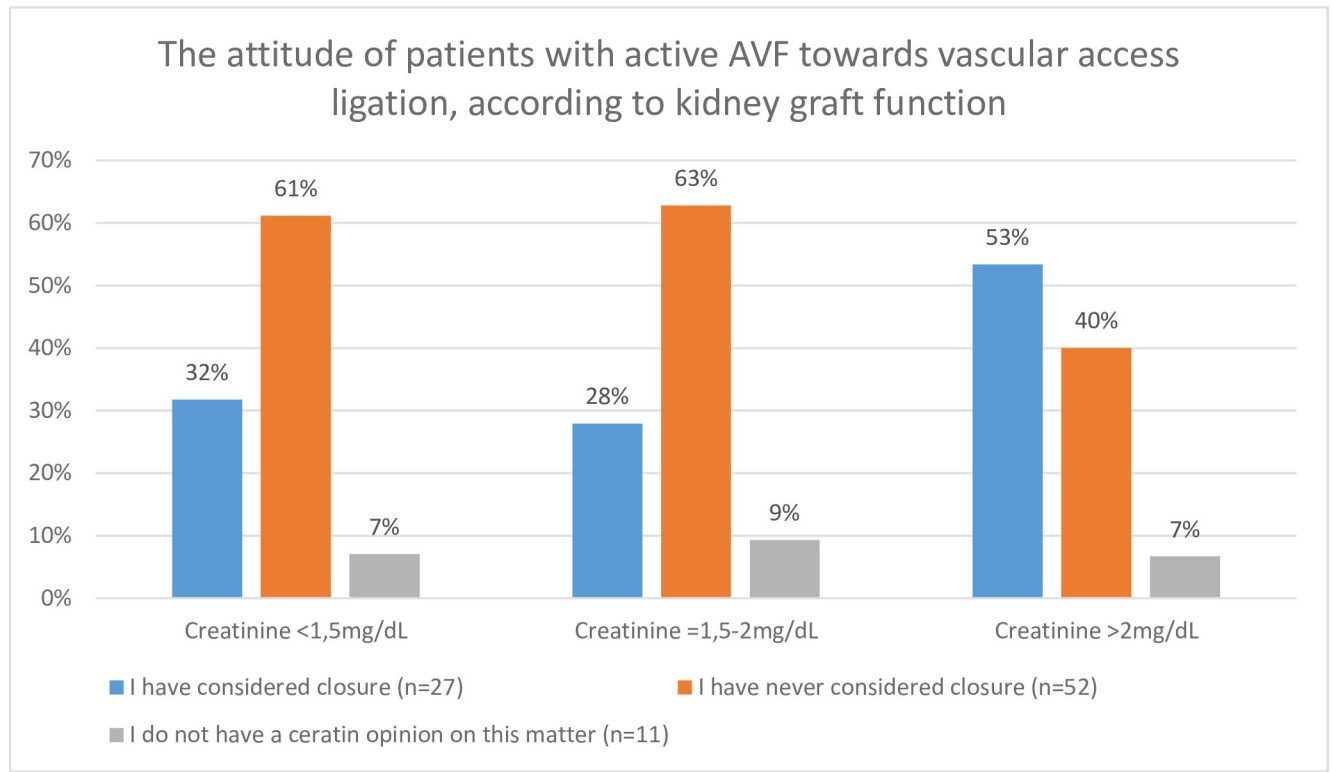

**Fig 1. Attitude of patients with active AVF towards vascular access ligation, according to kidney graft function.**

The proportions of patients who have considered AVF as an esthetic defect were similar in the groups of patients with forearm AVFs and more proximal vascular access (20/173, 11.6% and 6/65 9.23%, p = 0.6077, respectively). Among patients with active AVF, the highest proportion of those considering ligation was present in patients with forearm access compared to more proximal ones (Fig 3).

## Discussion

Many studies have already proven that the creation of AVF is a superior method of vascular access creation, as compared to arteriovenous graft (AVG) and central venous catheter (CVC). Among these three possibilities, AVF-patients have shown the lowest mortality, lower hospitalization rates, higher QOL and lower depression scores [11]. Among HD patients those with AVF have also shown the highest satisfaction with their vascular access [12].

Parallelly, the amount of kidney transplant recipients with persistent arterio-venous fistula rises systematically, due to better post-transplant care and development of transplantation programs. According to the guidelines of the European Society for Endovascular Surgery, the closure of a persistent vascular access after a successful kidney transplantation is not routinely recommended [13]. However, that indication is a Class I indication, Level of Evidence C, which means that it was predominantly formulated on the basis of the opinion of the expert group.

Recently, subsequent articles have been published, supporting the potential cardiovascular benefits of elective AVF ligation in patients with stable graft function. Firstly, the creation of vascular access in ESRD patients was associated with right ventricle (RV) dilatation, the incidence of which was in turn independently associated with an increased risk of death [14]. Secondly, the AVF-associated volume overload leads to left ventricle hypertrophy and cardiac

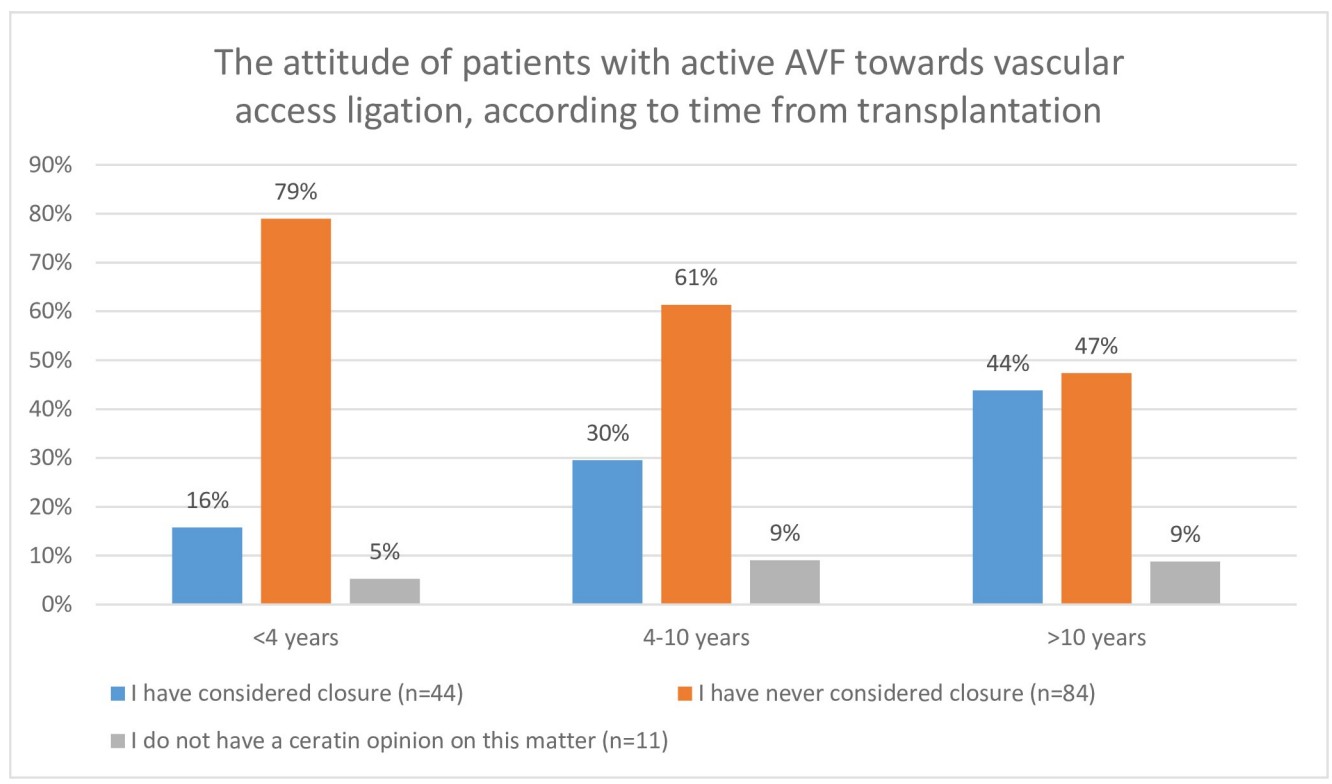

**Fig 2. Attitude of patients with active AVF towards vascular access ligation, according to time from transplantation.**

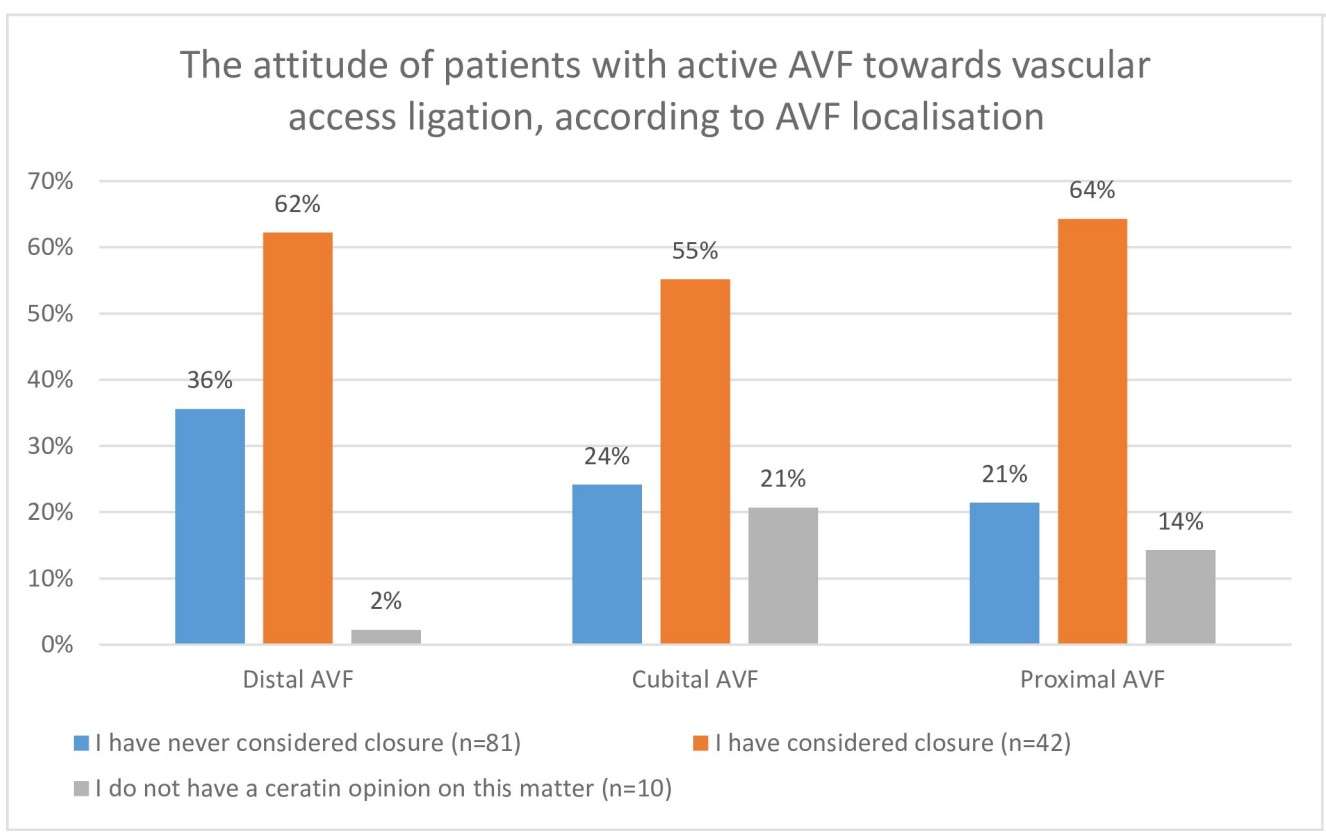

**Fig 3. Attitude of patients with active AVF towards vascular access ligation, according to AVF localization.**

remodeling [15]. Several studies have also presented a positive correlation concerning the AVF-flow, cardiac output and diastolic dysfunction severity, which is clearly an additional burden for patients with structural heart disease [16–18]. Some authors have also proven that such maladaptive changes (including RV dilatation) can be at least partially reversible after AVF-ligation or spontaneous thrombosis [14,19–22]. The reverse remodeling was expressed as a reduction of plasma NT-pro-BNP, reduction of left ventricle mass, left ventricle end-systolic volumes, left atrial volume and the improvement in the RV systolic function. The above observations have also been confirmed in a recently published randomized controlled trial, which is actually the strongest argument supporting elective AVF ligation [23].

However, no data on the long-term outcomes after such an intervention have been published yet.

Some studies also suggest that the localization of AVF in hemodialyzed patients may be associated with symptoms severity, which in turn influences the QOL [24]. These observations, however, were made in a cohort of dialyzed patients and it remains unclear whether they can be extrapolated from KTRs with persistent AVF.

On the other hand, there is almost no scientific proof supporting the thesis that AVF closure may be harmful. Weekers et al pointed out in their paper that the ligation of active AVF may be associated with accelerated decline of kidney graft function [25]. This observation has not yet been confirmed in any other publication. Other data have shown no association between kidney graft filtration function and the AVF ligation and its timing [26]. Parallelly, in an analysis of a large KTR's cohort, no all-cause mortality reduction has been demonstrated in subjects who had undergone AVF closure [27]. This study has, however, lacked

echocardiographic information and was limited to a three-year follow-up. Therefore, the main argument against such a procedure is undoubtedly the iatrogenic loss of vascular access, which must be recreated if the need for chronic dialysis occurs.

A recently published multi-center survey has also revealed that the opinions among experts regarding the post-transplant management of AVF show a considerable disagreement, especially regarding closure indications and qualifications. Moreover, a routine vascular access surveillance in kidney transplant recipients has been reported by 29% of the respondents [28].

However, there is no doubt that in order to achieve optimal treatment results a patient should be a part of a medical team himself/herself, which determines an optimal and personalized approach to the problem. Therefore, an extensive patient education, in terms of possible benefits and drawbacks of AVF ligation, is essential for a fruitful and trustful patient-physician cooperation.

In our study, 38.5% of patients did not give a clarified opinion on the topic of AVF ligation. Moreover, only 34.8% of the subjects who considered such intervention reported that their opinion or decision was prompted by an information acquired from their physician. Paradoxically, patients with active AVF and the worst renal graft function in the whole study group presented the highest proportion of subjects willing to ligate their vascular access (Fig 1). Thus, we alarm that there is a clear need for patients' education in order to raise awareness of the possible AVF management strategies, including not only ligation but also banding or flow-reduction [29,30], by providing thorough and evidence-based information source. In our study, the main source of knowledge of AVFs in the investigated patient cohort remains unspecified. Particularly interesting were the cases in which patients did not want to ligate AVF, despite the fact that they indicated AVF-related symptoms, as they were afraid of the potential return to dialysis. Although the decision making in such cases is extremely challenging, a proper patient education might be a key to achieving a reasonable consensus.

It also seems that patients with forearm AVFs tend to consider the AVF closure due to esthetic reasons more often than their counterparts with more proximal AVFs. However, these differences did not appear to be statistically significant in our study.

Taking account of the current knowledge and the current European Society for Vascular Surgery recommendations on the given matter, a post-transplantation routine AVF ligation cannot be implemented. In practice, it is reserved for patients with certain clinical complications, for instance: steal-syndrome, high output heart failure, infection or aneurysm formation, as these are related to significant mortality [27,31]. As a result, the role of a physician in the clinical decision-making regarding the management of AVF is limited, as the surgery qualification is partially patient-dependent and based predominantly on the occurrence of symptoms. Our study has not only shown that KTRs lack proper information about AVF ligation, but also that such a procedure is underutilized by underlying indications.

Our study has certain limitations. The questionnaire used is anonymous, and therefore all information provided in our study are self-reported. Moreover, the survey results could be modified by center-specific factors such as relatively low proportion of upper arm AVFs and reluctance to close asymptomatic AVFs in KTRs.

## Conclusions

The vast majority of KTRs have never considered AVF ligation and most of them did not receive medical assistance in the making of such decision.

There is a distinct need to raise patients' awareness in terms of post-transplant arteriovenous fistula management, so that kidney transplant recipients may actively and consciously

participate in the clinical decision making. We also strongly recommend a routine surveillance of persistent vascular access after transplantation.

## Supporting information

**S1 File. Questionnaire utilized in the study.**
(DOCX)

**S1 Data.**
(XLSX)

## Author Contributions

**Conceptualization:** Klaudia Bardowska, Krzysztof Letachowicz, Tomasz Królicki, Karolina Zajdel, Dariusz Janczak, Magdalena Krajewska.

**Data curation:** Klaudia Bardowska, Krzysztof Letachowicz, Dorota Kamińska, Mariusz Kusztal, Tomasz Gołębiowski, Tomasz Królicki, Karolina Zajdel, Oktawia Mazanowska.

**Formal analysis:** Klaudia Bardowska, Krzysztof Letachowicz, Tomasz Królicki, Karolina Zajdel.

**Investigation:** Klaudia Bardowska, Krzysztof Letachowicz, Dorota Kamińska, Mariusz Kusztal, Tomasz Gołębiowski, Tomasz Królicki, Karolina Zajdel, Oktawia Mazanowska.

**Methodology:** Klaudia Bardowska, Krzysztof Letachowicz, Tomasz Królicki, Karolina Zajdel.

**Visualization:** Tomasz Królicki.

**Writing – original draft:** Klaudia Bardowska, Krzysztof Letachowicz, Tomasz Królicki, Karolina Zajdel.

**Writing – review & editing:** Dorota Kamińska, Mariusz Kusztal, Tomasz Gołębiowski, Oktawia Mazanowska, Dariusz Janczak, Magdalena Krajewska.

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
