## [Decision Letter · Decision Letter 0]

15 Apr 2020

PONE-D-20-05938

The attitude of kidney transplant recipients towards elective arteriovenous fistula ligation

PLOS ONE

Dear MD PhD Letachowicz,

Thank you for submitting your manuscript to PLOS ONE. After careful consideration, we feel that it has merit but does not fully meet PLOS ONE’s publication criteria as it currently stands. Therefore, we invite you to submit a revised version of the manuscript that addresses the points raised during the review process.

ACADEMIC EDITOR: 

3 experts in the field showed interest in your work and I agree with them.

They came with a few additional questions and suggestions, and pleas revise the MS accordingly.

The MS will need quite a lot of work regarding English grammar and spelling by an official translator, so please take appropriate action (Editorial Office can advise).

We would appreciate receiving your revised manuscript by May 30 2020 11:59PM. To enhance the reproducibility of your results, we recommend that if applicable you deposit your laboratory protocols in protocols.io, where a protocol can be assigned its own identifier (DOI) such that it can be cited independently in the future. For instructions see: http://journals.plos.org/plosone/s/submission-guidelines#loc-laboratory-protocols

We look forward to receiving your revised manuscript.

Kind regards,

Frank JMF Dor, M.D., Ph.D., FEBS, FRCS

Academic Editor

PLOS ONE

2. Please include additional information regarding the survey or questionnaire used in the study and ensure that you have provided sufficient details that others could replicate the analyses. If you developed and/or translated a questionnaire as part of this study and it is not under a copyright license more restrictive than Creative Commons Attribution (CC-BY), please include a copy, in both the original language and English, as Supporting Information.

3. Please include in your Methods section the date ranges over which you recruited participants to this study.

5. Your ethics statement must appear in the Methods section of your manuscript. If your ethics statement is written in any section besides the Methods, please move it to the Methods section and delete it from any other section. Please also ensure that your ethics statement is included in your manuscript, as the ethics section of your online submission will not be published alongside your manuscript.

Reviewers' comments:

Reviewer's Responses to Questions

**Comments to the Author**

1. Is the manuscript technically sound, and do the data support the conclusions?

Reviewer #1: Partly

Reviewer #2: Yes

Reviewer #3: Partly

2. Has the statistical analysis been performed appropriately and rigorously? 

Reviewer #1: Yes

Reviewer #2: Yes

Reviewer #3: Yes

3. Have the authors made all data underlying the findings in their manuscript fully available?

Reviewer #1: Yes

Reviewer #2: Yes

Reviewer #3: Yes

4. Is the manuscript presented in an intelligible fashion and written in standard English?

Reviewer #1: Yes

Reviewer #2: No

Reviewer #3: No

5. Review Comments to the Author

Reviewer #1: The paper is performed following good research methodology, yet limited by the study cohort. The paper is well written well.

The impact of duration following transplant on patient's attitude towards AVF is essential to be made aware in the manuscript.

Reviewer #2: The paper is interesting and asks a very pertinent question.

There are some grammatical issues that will need correcting in the final proof.

I have two points:

1. Some of the AVF - patients have had their fistulas tied off. Does this not influence subsequent attitudes in the survey?

2. There are a predominance of distal fistulas in this group. Do you think this overrepresemnts the aesthetic issues? Where the forearm fistulas more likely to complain of aesthetic issues?

Reviewer #3: It is relevant clinical topic and hence the analysis is welcome. Authors are to be congratulated for undertaking it. The manuscript however needs a lot of revision from a language and grammar perspective

6. PLOS authors have the option to publish the peer review history of their article (what does this mean?). If published, this will include your full peer review and any attached files.

Reviewer #1: No

Reviewer #2: Yes: Nicholas Inston

Reviewer #3: No

---

## [Author Response · Author response to Decision Letter 0]

16 May 2020

Answer to Editor comments:

The manuscript was verified carefully to fulfill PLOS ONE requirements. Appropriate files are attached. The draft was checked and corrected by an official translator. The questionnaire used in the study was attached as a supplementary file in the original and English version. Time frames when study was performed were added to the manuscript. Raw data set is at-tached and was also added into repository. Information is provided in Methods section. Ethics statement was moved according to instructions. 

Answer to Reviewer 1 comments:

Thank you for your constructive remarks. The study was conducted in a limited number of patients from our center (about 30 %); however we got some interesting information that could be helpful in planning future trials. The approach to vascular access in a group of pa-tients after transplantation is a topic of growing interest. Differences in the approach to vascu-lar access in hemodialysis patients exist on country and international level, so there is a need to perform larger multicenter studies. The attitude to AVF closure in relation to time from transplantation was presented in revised manuscript. 

Answer to Reviewer 2 comments:

Thank you for your positive evaluation and pertinent questions. The survey results have to be analyzed in concordance with the practice of the center. Our approach is to create AVFs distally, majority of renal transplant recipients in our center have forearm fistula. Due to lower flows a lot of them occlude within few months from transplantation. We also do not consider forearm AVFs as cardiotoxic, that could be not true in some cases. AVFs are also ligated rare-ly. I am sure, that in different center the survey results could be quite different. Aesthetic issues are actually very subjective and difficult to measure. However it was the most common reason to consider or to close AVF, it is actually a bit surprising. We did not find the differ-ence in complaints profile in relation to vascular access location.

 Answer to Reviewer 3 comments:

Thank you for your positive evaluation. Language editing was performed.

---

## [Decision Letter · Decision Letter 1]

26 May 2020

PONE-D-20-05938R1

The attitude of kidney transplant recipients towards elective arteriovenous fistula ligation

PLOS ONE

Dear Dr. Letachowicz,

Thank you for submitting your manuscript to PLOS ONE. After careful consideration, we feel that it has merit but does not fully meet PLOS ONE’s publication criteria as it currently stands. Therefore, we invite you to submit a revised version of the manuscript that addresses the points raised during the review process.

ACADEMIC EDITOR:

Thank you for addressing the comments of the reviewers and modifying the manuscript accordingly. I would like to

Provisionally accept the paper, but it needs to be reviewed and revised by a native English speaker based on reviewers’ feedback.

We look forward to receiving your revised manuscript.

Kind regards,

Frank JMF Dor, M.D., Ph.D., FEBS, FRCS

Academic Editor

PLOS ONE

Reviewers' comments:

Reviewer's Responses to Questions

**Comments to the Author**

1. If the authors have adequately addressed your comments raised in a previous round of review and you feel that this manuscript is now acceptable for publication, you may indicate that here to bypass the “Comments to the Author” section, enter your conflict of interest statement in the “Confidential to Editor” section, and submit your "Accept" recommendation.

Reviewer #1: All comments have been addressed

Reviewer #2: All comments have been addressed

Reviewer #3: All comments have been addressed

2. Is the manuscript technically sound, and do the data support the conclusions?

Reviewer #1: Yes

Reviewer #2: (No Response)

Reviewer #3: Partly

3. Has the statistical analysis been performed appropriately and rigorously? 

Reviewer #1: Yes

Reviewer #2: Yes

Reviewer #3: Yes

4. Have the authors made all data underlying the findings in their manuscript fully available?

Reviewer #1: Yes

Reviewer #2: Yes

Reviewer #3: Yes

5. Is the manuscript presented in an intelligible fashion and written in standard English?

Reviewer #1: Yes

Reviewer #2: Yes

Reviewer #3: No

6. Review Comments to the Author

Reviewer #1: I am delighted for your efforts in performing a good clinical research project and well done to all involved.

Reviewer #2: Thank you for addressing the comments.

The paper is interestinga nd aconsiders a very valid subject that is understudied

Reviewer #3: The written English could be improved. The authors touch on a relevant subject which is if interest to a wide group of clinicians

7. PLOS authors have the option to publish the peer review history of their article (what does this mean?). If published, this will include your full peer review and any attached files.

Reviewer #1: No

Reviewer #2: No

Reviewer #3: No

---

## [Author Response · Author response to Decision Letter 1]

3 Jun 2020

Answer to Editor comments: The manuscript was reviewed and revised by native English speaker. 

Answer to Reviewer 1 comments: Thank you very much for your favorable review. 

Answer to Reviewer 2 comments: Thank you very much for your positive evaluation.

Answer to Reviewer 3 comments: The manuscript was reviewed and revised by native English speaker. Thank you very much for your constructive remarks.

---

## [Editor Report · Decision Letter 2]

5 Jun 2020

The attitude of kidney transplant recipients towards elective arteriovenous fistula ligation

PONE-D-20-05938R2

Dear Dr. Letachowicz,

We’re pleased to inform you that your manuscript has been judged scientifically suitable for publication and will be formally accepted for publication once it meets all outstanding technical requirements.

Kind regards,

Frank JMF Dor, M.D., Ph.D., FEBS, FRCS

Academic Editor

PLOS ONE
---

## [Editor Report · Acceptance letter]

22 Jun 2020

PONE-D-20-05938R2 

The attitude of kidney transplant recipients towards elective arteriovenous fistula ligation 

Dear Dr. Letachowicz:

I'm pleased to inform you that your manuscript has been deemed suitable for publication in PLOS ONE. Congratulations! Your manuscript is now with our production department. 

Kind regards, 

on behalf of

Dr. Frank JMF Dor 

Academic Editor

PLOS ONE